# A Novel Chip-Level Blockchain Security Solution for the Internet of Things Networks

**Hiroshi Watanabe** [1,*] **and Howie Fan** [2]

1   Department of Electrical and Computer Engineering, National Chiao Tung University,
    Hsinchu City 30010, Taiwan
2   3H Blockchain, Brea, CA 92821, USA; howie@sirforum.com
*   Correspondence: hwhpnabe@gmail.com; Tel.: +886-3-5731704

**Abstract:** The widespread computer network has been changing drastically and substantially since blockchain and IoT entered the stage. Blockchain is good at protecting data transactions between logical nodes with a desirable guaranty. Internet of Things (IoT), on the other hand, by providing ultimate convenience to consumers, is expected to give rise to many various merits in a broad business scene. The security of IoT is still an open problem and if blockchain can reinforce IoT security, as many authors have hoped in recent papers, these newcomers appear to make a good collaboration to reinforce IoT security. However, software copes with logical nodes and IoT involves a vast number of physical nodes (IoT devices). Enabling blockchain to protect IoT cannot be brought to reality without respectively identifying logical and physical nodes. This is identical to the Proof-of-Trust problem. In this article, we propose a conceptual solution—Blockchained IoT—and show that this concept is able to be realized on-chip level using mass-produced dynamical random access memory (DRAM). We have completed the first test of longevity and temperature dependence ($-40\ ^\circ$C to 105 $^\circ$C) to confirm the necessary characteristics for the 5G base stations that are known to have an issue of self-heating. Furthermore, we have coarsely evaluated the probability of two DRAM IC chips being associated with an identical cyber-physical chip identification accidentally. Then, such a probability is minimal.

**Keywords:** blockchain; IoT; security; identification; authentication; connected devices; spoofing; cyber-attack; Proof-of-Trust; DRAM; IC chip

## 1. Introduction

The Internet-of-Things (IoT) is expected to change the rules of the game for the industrial eco-system thanks to its big potential to provide unprecedented convenience to customers, to substantially improve the efficiency of the usage of social infrastructures and to drastically increase data acquisition for Big Data. Such prospects may come true as IoT devices will get widely adopted and promote the efficiency of artificial intelligence. In other words, with an increasing number of IoT devices deployed all over the cyber network, IoT is likely to become the next disruptive technology. In recent years, the majority of connected devices is switching from computers to various smaller devices; it is expected that in 2020 there will be 5–10 devices per person on average [1]. In this way by 2020 the number of connected devices could be drastically increased to reach 26–50 billion [1,2]. Most end-users may not manage all of them in a secure way. This technology may be used to collect an astonishing amount and variety of information with or without end users' prior consents. One who will collect much amount of such an information may dominate and control, as a ruler, the communications on the IoT network. Many computer scientists have pointed out the risk and damage of cyber-attacks as the IoT network grows [3–10]. There are three kinds of security concerns; Firstly, it is that the

ruler can control everyone's information including a personal privacy. Secondly, it is nevertheless difficult for the ruler to always protect everyone's information from advanced cyber-attack on the IoT network. Finally, the growth of the IoT network extends the field of damage from the cyber-world to the real-world such as: shutdowns and runaways of mass-transport systems, large-scale blackouts, factory shutdowns and so forth. It is important to realize that IoT securely connects the cyber-world and the real-world.

In general, there are four ways to protect information. The first one is the local data protection, meaning that data stored in a local storage is protected by a sufficiently strong encryption. This is workable even outside of the network itself by using symmetric key cryptosystem. The random number generator (RNG) [11–13] is able to enforce this. However, the state of affairs changes drastically in the cyber-networks, because data protection offers less design flexibility to avoid a certain type of cyber-attacks. In particular, no matter how strong the encryption is, it is useless against a novel ransomware cyber-attack [14]. The ransomware attack does not break the encryption of target data and may encrypt the target data afresh. The user or owner of the target data will be required to pay ransom money to receive the encryption key to unlock the data. Moreover, in the case, where the adversary aims to disturb the usage of the data, the attack is complete solely by encrypting the data. The second data protection approach is the central security. In this case, data stored in a local node (client) is monitored and protected by a central node (server) securely connected to the local node. For example, a big communication company that holds a central server may protect clients' data in a client-server (CS) network. Nonetheless, the information leakage of users' data is recently often reported in the media [15]. The cyber-attack does not require human's assistance and can be repeated endlessly, frequently and simultaneously all over the world. This implies that a cyber-attack may succeed somewhere at last even though the probability of an adversary winning each time is less than 0.001%. Moreover, there is a countless number of adversaries all over the world and most of them are anonymous, while the number of central servers is relatively limited. In addition, the technique of cyber-attacks are constantly evolving all the time and adding in complexity, each year becoming more and more difficult to fight off. Third information protection method is so-called Cloud/edge security. In which case, instead of being stored in a local node (clients' platform), data resides at the cloud servers equipped with advanced security protocols and at the remote mirroring storages. If necessary, cloud storages and edge servers can be accessed by a client from the correspondent local node via virtual private network (VPN). In the case, when the data was irregularly altered, or the storage was damaged, the server can restore the original copy from the mirroring node without service interruptions. This method is also effective against the ransomware attacks. However, no matter how low the adversaries' chances to penetrate security barriers are, each moment every server must be able to withstand countless repetitions of all kinds of cyber-attacks. In this way, cloud/edge storage is similar to the central security. Besides, such a pressure may significantly increase the maintenance cost of edge computing. It may be unrealistic to consider that all edge servers all over the world can be eternally supplied with sufficient budget and human resources and can be always set up with latest launched hardware and software. An invaded server, after losing the battle, may start working for the adversary from an inside of the authorized servers' network. This server may now function as an intermediate device to store and route data flow between two secure devices (the so-called "attack of man in the middle") [16]. Thus, the adversary can collect communication information via this server. Even though network might be equipped with homomorphic encryption, the adversary may intercept the communication and send false signals between two secure devices. It is self-evident that the central security and cloud/edge security are superior techniques of information protection comparing to the data protection in the network. However, the number of nodes in the IoT network is vast and this is posing limitation on the capabilities of the central security and the cloud/edge security to protect the customers' digital assets and privacy from all cyber-attacks. The fourth technique of information protection is related to the monitoring of communication, in which case data transaction itself is examined for early detection of evasive malware execution. This approach makes it easier to

counteract against new developing malware [14]. The monitoring itself appears not for protecting customers' data; but if we consider saving the monitoring records, then this idea can be expanded to lead to a new concept, namely, the registration of a batch of communication records in distributed ledgers. In blockchain, a batch of data transaction records is registered in a block in the blockchain, which records are jointly protected by all participating nodes by a method like the majority decision. The blockchain security is likely to become more powerful as the number of participating nodes increases. This feature is beneficial in IoT because in IoT the number of nodes has the tendency to increase explosively. In recent years, it is expected that blockchain can enforce the security of IoT network [17–31]. Furthermore, a significant merit of blockchain is to distribute the maintenance cost of edge servers. This is especially beneficial to protect weak servers suffering from a lack of security resources.

The remaining issue is the joint consistency of IoT and blockchain [28–31]. It is well-known that IoT is heterogeneous. Nodes in IoT are represented by various hardware entities such as smart phones, routers, wireless base stations, connected cars, smart houses, industrial factory machines and deployed sensors for the analysis of agriculture and environment and so forth. Thus, their functions and computational power are inherently different (that is, heterogeneous). Usually, deployed sensors are the constrained devices that have less storage and less computational power and they communicate with the rest of the network at low-rate wireless frequencies. On the other hand, servers, routers, base stations and other smart devices usually are given sufficient storage and power of frequent arrhythmic execution for luxury protection. That is why the protocol has to be designed taking into account all the properties and limitations of all of heterogeneous physical nodes [3]. An adversary may find a weak physical node whose security resource is temporarily low and then spoof it. Besides the accounts in blockchain are allocated with public keys and the public keys are logical addresses of logical nodes having no heterogeneity generated from the individuality of hardware entities serving as physical nodes. Therefore, we are required to link physical nodes having the physical substance to logical nodes having no physical substance. Here we face a new interdisciplinary problem between hardware and software, which makes it difficult for the Proof-of-Trust to be incorporated in the IoT network. Therefore, it is difficult to resolve it solely by the software improvement or solely by the hardware improvement. It is discussed in Reference [26] "*How can semiconductor companies help resolve security issues in the IoT network, which is the greatest obstacle to growth of the IoT business.*" From the viewpoint of industrial and end users, IC chips should be designed to have anti-tamper traceability of dataflow in the IoT network. In this article, we propose a conceptual solution to this problem and then demonstrate that it is an exemplary application.

In Section 2, we briefly review the concept of blockchain, which may be helpful to understand our concept. In Section 3, we explain the proposed solution to combine blockchain and IoT, that is, Blockchained IoT (BIoT). Sections 4 and 5 are devoted to the discussion and conclusion, respectively.

## 2. Brief Review of Blockchain

As illustrated in Figure 1a, there is a plurality of logical nodes (depicted by circles) respectively allocated with logical addresses in the logical layer-1. This logical layer-1 is established on the data link layer. The data link layer is established on the physical layer at the bottom. In the logical layer-1, the communication (data transaction) is executed between logical addresses, while it is assumed to be protected by the security systeM−1 therein. Now, suppose that the security systeM−1 is broken. In this case, defenders are forced to fix vulnerability and then establish the logical layer-2 to be protected by the security systeM−2, as illustrated in Figure 1b. If the security syteM−2 is subsequently broken, the logical layer-3, protected by the security systeM−3, is established on the logical layer-2, as illustrated in Figure 1c. Consequently, the logical layers have been repeatedly updated above the datalink layer to fix the vulnerabilities.

There was a great breakthrough in 2008, with invention of the blockchain that plays a central role of the bitcoin technology [32]. It has allowed cryptocurrency to be sent directly from one party to

another without going through a financial institution. The bitcoin addresses serve as logical addresses on the logical network (i.e., public keys). As illustrated in Figure 1d, the data transaction among those logical addresses is practically-strictly protected from the illegal falsification, as long as blockchain is tough enough. Thereby, blockchain has ignited the advancement in financial technology in recent years [33]. On the other hand, as shown in Figure 1a–d, there is a plurality of physical nodes (depicted by squares) respectively allocated with physical addresses in the physical layer. Since data (e.g., a frame) is transferred among those physical addresses in IoT, blockchain has to identify the physical addresses on the network to protect the communication. Unfortunately, physical and logical addresses are not connected to each other, as illustrated in Figure 1a–d. Therefore, neither blockchain nor the repeatedly updated security systems can protect data transaction between physical addresses. Nevertheless, the IoT is the physical network comprising a huge number of connected devices (physical nodes) respectively allocated with physical addresses. Then, a practical solution to the IoT security problem is to combine physical addresses with logical addresses, respectively, in a way that is fully blockchain-compatible at smallest possible penalty.

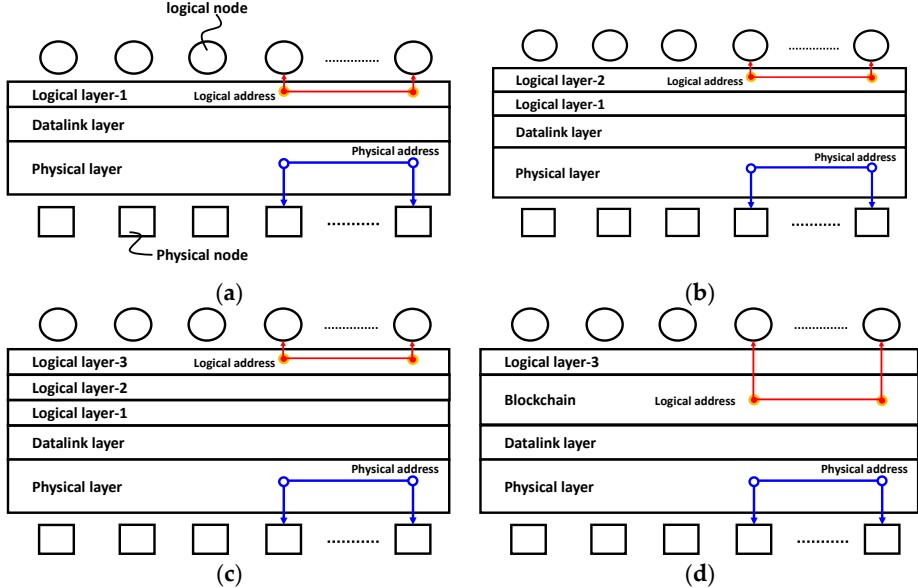

**Figure 1.** Communication layers: (**a**) the logical layer-1 above the data link layer; (**b**) The logical layer-2 is laminated on the logical layer-1; (**c**) the logical layer-3 is laminated on the logical layer-2; (**d**) the blockchain is inserted between the logical layer-3 and the data link layer. Each new layer appears as security layer because the previous one was compromised.

## 2.1. Transaction

In Figure 2, we illustrate a sequence of transactions among three logical nodes. In each transaction, digital data is sent from the sender's logical node to the receiver's one. The bulk squares at the bottom are the transaction units, each of which has a public key, a hash value and an electronic signature. Each logical node is composed of a transaction unit and a secret key. The public keys create unique pairs with the secret keys. As mentioned above, the public keys are the logical addresses of the logical nodes. In the sender's logical node, the hash value is generated by hashing the transaction unit of the sender's logical node with a hash function (e.g., SHA-256) and then transferred to the receiver's logical node while including the sender's logical address in the hash value. In the sender's logical node, the electronic signature is also generated by encrypting the receiver's public key and the generated hash value by using the sender's secret key and then forwarded to the receiver's logical node. In the receiver's logical node, we can decrypt this received electronic signature by using the sender's public key and then check if it coincides with a set of the receiver's public key and the received hash value.

In this way, we can confirm that the received hash value has been indeed forwarded from the sender's logical node.

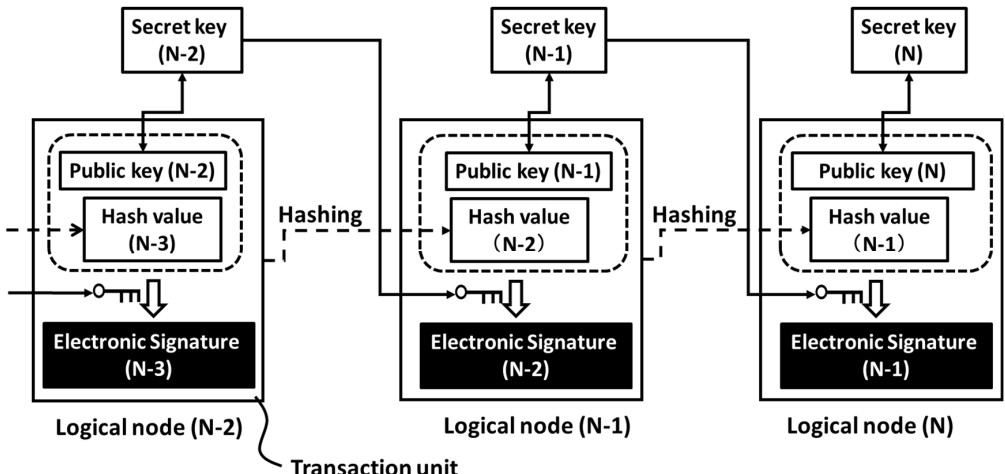

**Figure 2.** Data transaction between logical nodes. Dashed arrows denote the hashing.

In other words, the hash value (N−1) has been transferred from the sender's logical nodes (N−1) to the receiver's logical node (N) while including the sender's logical address (N−1) with the certificate of the previous sender's electronic signature (N−2). The hash value (N−2) has been transferred from the sender's logical nodes (N−2) to the receiver's logical node (N−1) while including the sender's logical address (N−2) with the certificate of the previous sender's electronic signature (N−3). The hash value (N−3) has been transferred from the sender's logical nodes (N−3) to the receiver's logical node (N−2) while including the sender's logical address (N−3) with the certificate of the previous sender's electronic signature (N−4) and so forth. At last, we can find that the logical node (N) saves the hash value (N−1) including all past senders' logical addresses (N−1), (N−2), (N−3) and so on and the certificate of all past previous sender's electronic signatures (N−2), (N−3), (N−4) and so on. This is the signed record of the past transactions. Note that, in order to generate an electronic signature, an encrypting key is a secret key and a decrypting key is a public key (i.e., logical address).

*2.2. Generation of Blockchain*

The records of past transactions signed by an electronic signature are included into the latest hash value. However, those signed records are still vulnerable if sooner or later an adversary will acquire the ability to break the encryption or hash function. To fix this, in blockchain, a batch of signed records is registered by the consensus algorithms such as Proof-of-Work (PoW), Proof-of-State (PoS), Proof-of-Importance (PoI), Proof-of-Consensus (PoC) and so forth. Bitcoin adopts PoW, whereas PoS is adopted in Ethereum. The PoI and PoC are used in NEM and Ripple, respectively. To this moment, all listed consensus algorithms have been tested and validated. Let us take PoW as an example to how the consensus in blockchain is obtained (registered).

In general, a plurality of logical nodes is able to transfer data to the latest logical node. Then, the trajectory of transferring record may form a tree diagram (called "Merkle's tree") [34], as illustrated in Figure 3. Each dashed arrow corresponds to data transfer illustrated in Figure 2. In this example, there is the latest one at the bottom, which has come from three hash values A, B and C. The hash value A has come from the hash values A1 and A2. The hash value A2 has come from the hash values A21 and A22 and so forth. No matter how complicated the tree diagram is, there is only the latest hash value at the bottom (called "Root of Merkle"). This can be the representative of all hash values which are included into the same Merkle's tree. Usually, several hundred transactions are bunched to form a block labeled by a Root of Merkle. As illustrated in Figure 4, the block (M−1) comprises the Root of Merkle (M−1), the nonce (M−1) and the block hash (M−2). The block hash (M−1) is

generated by hashing the block (M−1) by tuning the nonce (M−1) to satisfy the linkage requirement that the first 16 digits are all zeros in the generated block hash (M−1). At first, a nonce value is given as an initial candidate for the nonce (M−1). Then, we check if the first 16 digits of block hash to be generated using the given nonce value are equal to all zeros. If not, a different nonce value is given. Then, we check if the first 16 digits of block hash generated using the updated nonce value are all zeros. We repeat this procedure until the first 16 digits of block hash become all zeros. The nonce (M−1) is the finally selected nonce value. Since the hash function is irreversible, this tuning requires significant computational power. The block hash (M−1) is then publicized on the network to register the block (M−1). Note that the block (M−1) is linked to the previous block (M−2) by tuning the nonce (M−1). Suppose that a user finds a new bunch of transactions that have not been registered yet, which are labeled by the Root of Merkle (M). If the user succeeds in tuning the nonce (M) to satisfy the linkage requirement, then he or she can register the new block (M) and link it to the block (M−1). In PoW consensus algorithm, this user is referred as a *MINER* and registration algorithm is called *MINING*. By repeating the mining of new blocks from the past to the future, the blockchain is formed, as illustrated in Figure 5. The dotted square corresponds to the linkage in Figure 4.

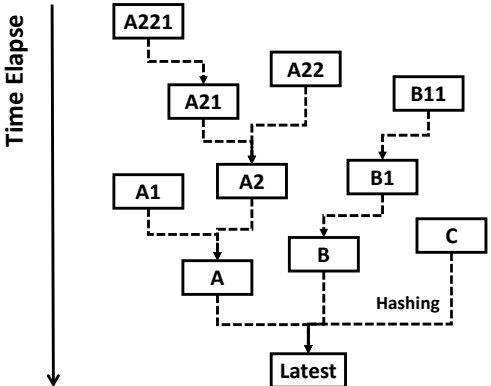

**Figure 3.** Merkle's Tree, wherein the dashed arrows denote the hashing transactions. "Latest" at the bottom is the Root of Merkle.

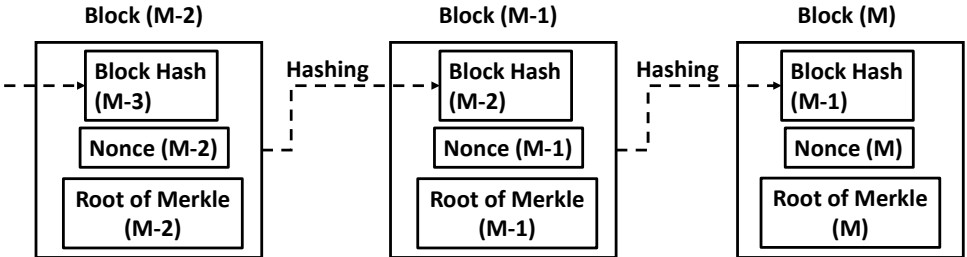

**Figure 4.** Block-to-Block linkage, wherein the dashed arrows denote the hashing.

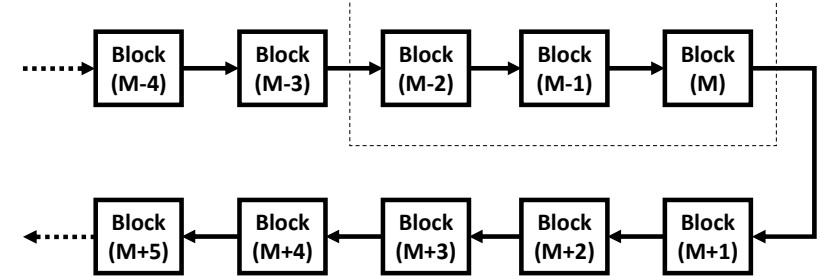

**Figure 5.** Blockchain, wherein the dotted square corresponds to the linkage in Figure 4.

## 2.3. The Length of Blockchain and the Strrength of Anti-Tampering

Suppose that an adversary tries to alter data related to the hash value A2 in Figure 3. This tampering influences the hash value A, because it includes the record of the past transaction. In other words, the manipulation in A2 changes the transaction record until A; and then the hash value A is altered. If A is manipulated like this, the hash value "Latest" (at the bottom of Figure 3) is influenced in a similar manner. This means that tampering a part of the transaction record will change the Root of Merkle. Suppose that the Root of Merkle (M−1) is altered by this tampering in Figure 4. This breaks the linkage requirement that the first 16 digits are all zero in the block hash (M−1). In order to recover the linkage, the adversary is forced to revise the nonce (M−1) to recover the linkage requirement from the block (M−1) to (M). However, the change of the nonce (M−1) will change the remaining digits of the block hash (M−1) and then the linkage of block (M) to (M+1) is indeed broken. To recover this linkage, the adversary is further forced to revise the nonce (M). Therefore, in order to leave no trace of the tampering in A2, this adversary must revise the nonce (M−1), the nonce (M), the nonce (M+1), the nonce (M+2) and so forth. Consequently, as the length of blockchain is increased, more computational power is consumed to tamper a part of the transaction record. Therefore, the strength of anti-tampering of transaction record is enhanced as the blockchain is elongated.

## 2.4. Limitation of Blockchain Protection

All logical nodes have logical addresses (i.e., public keys) and the data transaction between those logical addresses can be protected by blockchain, as explained above. Each software application running at the blockchain infrastructure is allocated with the logical address correspondent to the application account. In a similar manner to the public key encryption, the public keys are uniquely coupled to the secret keys. As long as the secret keys are protected, the blockchain can prevent the illegal falsification of data transaction between the logical addresses. However, the transaction in IoT is executed between physical addresses which correspond to connected devices (hardware). Physical and logical addresses are not uniquely combined, as illustrated in Figure 1a–d. In other words, an adversary may thieve a physical address (i.e., spoofing) to eavesdrop data exchanged between other two connected devices ("man in middle attack" [16]). This is the limitation of the existing blockchain technology, confirming its ability for the IoT network protection.

## 3. Blockchained IoT (BIoT)

Suppose that a physical node on the cyber network is allocated with a logical address serving as a public key used in the data transaction illustrated in Figure 2. This physical node may be either of the connected devices such as a router, a connected car, a healthcare sensor of diagnostic, a 5G base station and so forth. It appears that if blockchain protects the data transaction of this logical address with other logical addresses, we can protect the data transaction of this physical node with other physical nodes. However, blockchain cannot distinguish a kind of device which the logical address is allocated to. Is it a drone, an autonomous car or a 5G base station? One might think that it is possible to link a logical address to a physical address (e.g., MAC address) allocated to a connected devices. However, the spoofing of physical address is a desirable target for an adversary. We should note that the reallocation of a logical address has been used to replace an old hardware with a new one in the authorized network. Indeed, an adversary can reallocate the logical address installed to a physical node to another physical node while not altering the data transaction record protected by blockchain. If an adversary reallocates a logical address of a 5G base station to his or her own computer (i.e., the spoofing of a physical node), he or she can eavesdrop data transaction between the other two authorized nodes, where the data transaction is protected by blockchain. Even if the network was protected by homomorphic encryption, adversary can replace regular communication signals with fake ones and shut down the communication. Besides, the spoofing of physical address enables an adversary to take over control of the machines. In order to make sure that the linkage of a physical

node to a logical account cannot be compromised, we propose a new method to connect a smallest component of a physical node to a logical address with no chance of an adversary's reallocation.

*3.1. On a Chip-Level Countermeasure*

Note that semiconductor chips are the smallest components of connected devices (i.e., physical nodes) and that no adversaries can reallocate a chip from a connected device to another using software intervention through the internet. Accordingly, defining a physical address of the connected device has to be required to be equivalent to identifying the semiconductor chip included in the connected device, on the cyber network. This is exactly what the cyber-physical chip identification is about. A practical implementation of the cyber-physical chip identification is to extract a physically unique specification from a mass-produced semiconductor chip of a connected device. In a constrained device that has less storage, computational power and communicating with others at low-rate wireless, we can extract a physically unique specification from the device's embedded memory. In a device that has sufficient storage and computational power, we may extract a physically unique specification from an equipped memory chip. In this article, we discuss the latter, that is, the extraction of the cyber-physical chip identification from mass-produced memory chips.

3.1.1. Existing Solutions

Someone might suppose that a cyber-physical chip identification can be realized by an integrated Random Number Generator (RNG) [11–13] or by an on-chip Physically-Unclonable Function (PUF) [35–38].

RNG

In the usage of integrated RNG, we can combine a random number with a chip identification code in order to encrypt it. To decrypt the encrypted chip identification code, we may subtract the combined random number from the encrypted one. In this way, only the one who knows this random number can encrypt and decrypt the chip identification code. Like this, the random number serves as both an encrypting and decrypting keys, which is a symmetric key cryptosystem (i.e., secret key = public key). Now assume that an adversary sends a plain text to a connected device and then obtains a response encrypted by a random number generated inside the connected device (encrypted text). The adversary can repeat this procedure to get a plurality of sets of encrypted and plain texts. If random numbers used to generate those encrypted texts are not truly random (i.e., pseudo-random), the adversary can then reproduce the algorithm that was used to generate those pseudo-random numbers. Therefore, we should encrypt chip identification codes using physical (true) random numbers and then "securely" store those random numbers inside IC chips since later on they are needed to decrypt chip identification code. However, to exchange messages, the random numbers (symmetric keys) must be forwarded to the recipients in advance. There are no methods to transfer symmetric keys securely via the network. This is the main reason, why public key cryptosystem was invented [39].

PUF

In the usage of the on-chip PUF, a connected device includes a special circuit which uniquely responds to an input. If the operational function of this special circuit is physically unclonable, then it is a PUF circuit. If we have a PUF circuit inside a chip, we input a chip identification code as well as an input code from the external of the chip to the inside once. Inside the chip, there is an activation code constructor, a PUF circuit and an ID generator. The PUF circuit receives the input code to generate a response. This response and the input chip identification code together are forwarded to the activation code constructor, which in turns generates an activation code inside the chip. We may store this activation code inside the chip and delete the chip identification code. When we need the chip identification code for online usage, we may input the input code from the external to the chip again. This input code and the stored activation code are then forwarded to the ID generator to

reproduce the chip identification code inside the chip once again. After signing-in, we may delete the chip identification code again. In this way, the PUF can hide the chip identification code from the external access. The PUF enables connected devices with an authenticated chip to use online service. However, this is different from protecting data transaction among connected devices.

Besides, we should note that the stable shipment in large quantity is indispensable to cover the entire or the majority of IoT network while satisfying the mass-product specifications in Table 1 [40]. It is unfortunately difficult for an existing on-chip PUF to satisfy this requirement. For example, it is reported that data turnover occurs in 30% of bits of encryption data as temperature raises from 25 degree Celsius (°C) to 85 °C [41]. Therefore, the temperature sensor is installed on the chip to prevent the output of the encryption data from the chip while atmospheric temperature is out of the predetermined range [41,42].

**Table 1.** Demanded specification for Internet of Things (IoT) chips.

|  | **Automotive** | **Industry** | **Consumer** |
| --- | --- | --- | --- |
| Longevity | >10 years | 5 to 10 years | 1 to 3 years |
| Temperature | −40 to 150 °C | −40 to 85 °C | 0 to 70 °C |
| Humidity | 0–100% | High | Low |
| Failure Rate (ppm) | As low as possible | <100 | <300 |

### 3.1.2. Cyber-physical Chip Identification

The smallest component of connected devices is a semiconductor chip, as mentioned above. In order to protect the malicious reallocation of physical entity without using PUF, *we must confine a physical address inside a semiconductor chip mounted into our connected device even while exchanging messages with the other devices on the network.* We can propose a potential solution by using the public key cryptosystem, where the public key serves as a logical address to exchange messages with others and a secret key is hidden in and *"uniquely" allocated to* a semiconductor chip inside the connected device. Since we must avoid the accidental duplication in the allocation of secret keys as well, the cyber-physical chip identification must be able to contain large quantity of information. Additionally, in order to suppress the increase of manufacturing cost of mass-produced semiconductor chips due to the implementation of cyber-physical chip identification, we have to avoid any changes in the front-end process in the chip manufacturing and minimize any penalty in chip area. An example of the cyber-physical chip identification under discussion has been demonstrated by using manufactured semiconductor chips of dynamical random-access memory (DRAM) chip [28–31]. In general, DRAM memory cells are arranged in a two-dimensional checker board array expanded by bit and word lines. Each memory cell is located at each cross-point of bit and word lines. The cross-points can be allocated with addresses by corresponding bit and word lines, respectively. Some of the memory cells are irreversibly defective, called failure bits and their addresses are identified during the test before shipment. The failure bits are generated in the manufacturing process. The generation of failure bits is out of control. Therefore, the failure bit generation can be considered physically random. Accordingly, a mass-produced memory chip has a physically random distribution of failure bits on its memory cell array, unless the failure rate is zero. Suppose that the total number of memory cells is four-billion and the number of failure bits is one thousand. This corresponds to the failure bit rate 0.000025% and the bit capacity of chip 4 Gb. As long as the failure bit distribution is totally random, the number of the failure bit combinations is about $2.85 \times 10^{7034}$. Even if there could be 100 trillion cyber-physical nodes in the entire cyber-network, the probability that two chips are accidentally allocated with an identical cyber-physical chip identification is about $3.5 \times 10^{-7021}$.

### 3.1.3. Experiment

Figure 6 shows a part of the cyber-physical chip identification code that we have successfully extracted from a mass-produced chip of 1Gb DDR1 DRAM. In order to check the longevity listed in

Table 1. we apply the heating acceleration test. This test is typically used to characterize the data retention of non-volatile memory chips. The condition of 125 °C for 168 h corresponds to ten years' data retention, which is the strictest longevity criterion in Table 1. We have adopted this condition to perform ten years longevity test of the cyber-physical chip identifications of 1,116 mass-produced chips of 1 Gb DDR1 DRAM as illustrated in Figure 7 and following the guidelines from [28–31]. First, we have measured the identification codes generated from those 1116 chips. Next, the 1116 chips were heated at 125 °C for 168 h (bake). Then, we have measured the identification codes generated from the 1116 chips again. In Figure 8a,b, we show parts of the cyber-physical chip identifications before and after the bake of the chip number 11 (a) and 606 (b) among those 1116 sample chips. All bits are consistent before and after bake. In Figure 9, we show the statistics of this measurement. As a result, we found that all 1,116 chips have no inconsistent bits before and after the heating accelerated test. This is a consequence of the fact that the failure bits are irreversibly defective. Hence, we validated the specification of ten years of longevity by using 1116 chips of mass-produced chips of 1Gb DDR1 DRAM.

We also performed the temperature specification test by using 124 chips of 1 Gb DDR1 DRAM as illustrated in Figure 10 and following the guideline [28]. Firstly, we have recorded the identification codes generated from a chip of 1 Gb DDR1 DRAM in turns at 27 °C, 105 °C and −40 °C. Then, we check if those codes are consistent in all bits at all temperatures. We have repeated this procedure for 124 different chips. In Figure 11a,b, we show parts of the cyber-physical chip identifications measured at 27 °C, 105 °C and −40 °C from the left to right of chip number 16 (a) and 121 (b) among 124 sample chips. All bits are consistent at all temperatures, respectively. In Figure 12, we show the statistics of this measurement. As a result, it is found that all 124 chips have no bit flips in the temperature range from −40 °C to 105 °C. This is attributable to the fact that the failure bits are irreversibly defective. It is thus confirmed that our cyber-physical chip identification can stably work at least in the temperature range from −40 °C to 105 °C, with no doubt. This is crucially important for the IoT security implementation for a 5G base station, which commonly experiences the self-heating due to the high-speed communication. In addition, since the cyber-physical chip identification is resistant to temperature changes like this, we do not need to embed a temperature sensor into a chip for the counterplan to a fault attack. This implies no penalty in chip area even though installing the cyber-physical chip identification into a mass-produced memory chips. The temperature independence is an important advantage of the cyber-physical identification. The changes in humidity also will not change the failure bit distribution either, as long as the failure bits are irreversibly defective. In other words, we may choose a failure mode to satisfy requirements selected from Table 1 at the pre-shipment test. If we evaluate this randomness by using the redundancy columns of 4Gb DRAM, the probability that two chips are accidentally allocated with an identical cyber-physical chip identification is about $1 \times 10^{-2620}$, which is still negligibly small. Moreover, according to Moore's rule, we can expect that this probability will be successfully reduced. If the failure bit distribution is dependent of some factor, the distribution then may be compounded with the identification code regarding that particular factor. Examples of those identification codes may be such as the identification codes of the factory, the manufacturing date, the production line and so forth.

We should note that the generation of the failure bit distribution inside a memory chip comes with no penalty in chip area and zero additional front-end-process steps. A pair of secret and public keys can be generated from the cyber-physical chip identification code, in a way that is described in the next subsection.

```
00000001101001111100000000001101011011100000000001110001000000000000000111011011110
00000001000000010100100000001000001011111000000010001100011110000000100011100100
00000000100011110011100000001001001110100000000001001011011100000000010011001101
01000000011011100100000000000110111111010000000000111000000001100000001110100000
01000000000111010000001100000010000011010001000000010000101000101000001000010101
01010000001001001100011000000010010011111001000000010011101110101000000100111111
11110000000010100100000000000000101011100111100000000101011110001010000001011001
10101100000010110011101010000000101110011010100000001100000011001100000001100010
10101110000001101010000000000000110110010010100000001101110011010100000011011
10101100000000111001111001010000000111100000010100000011110100000001000000011110
11000110100000001111001001011000000111110101100110000001111101101110100000001111
11100010100000000011010011110000000001101011011100000000011100010000000000000001
11011011110000000010000001010010000000010000010111110000000100011000111100000001
0001110010000000000010001110011100000001001001110100000000010010110111000000000
10011001101010000000110111001000000000001101111110100000000111000000001100000
01110100000010000000111010000001100000010000011010001000001000010100010100000
01000010101010100000010010011000110000000100100111110010000001001110111101010000
00100111111111100000001010010000000000001010111001110000000101011110001010000
00010110001101011000000101100111010100000010111001101010000000110000011001100
00001100010101011100000011010100000000000011011001001010000000110111001101010
00000110111110101100000001110011110010100000011100000010100000011110100000001
000000011110110001101000000011110010010110000001111101011001100000011111011011110
1000000111..................
```

**Figure 6.** A part of generated cyber-physical chip identification code [28]. This was extracted from a mass-produced memory chip of 1Gb DDR1 DRAM. We can regard that in fact different chips exhibit unequal cyber-physical chip identification codes.

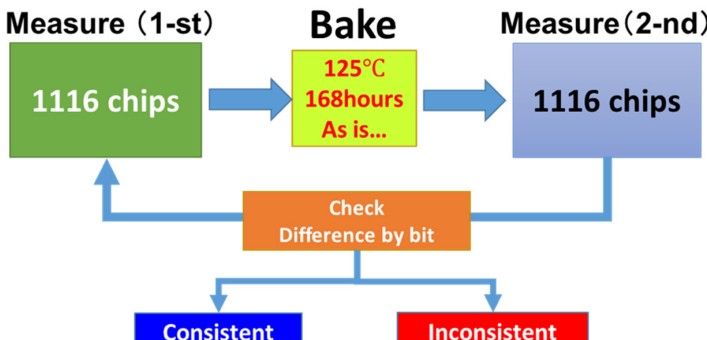

**Figure 7.** A drawing to illustrate the procedure of longevity test. The 1116 sample chips are the mass-produced 1Gb DDR1 DRAM IC chips.

Before Bake

After Bake

**(a)**

**Figure 8.** *Cont.*

Before Bake

After Bake

(**b**)

**Figure 8.** A part of the cyber-physical chip identifications before bake (**left**) and after bake (**right**) of chip number 11 (**a**) and 606 (**b**) among 1116 sample chips. All bits are consistent before and after the bake, respectively. This shows that our cyber-physical chip identification must be, with no doubt, resistant to temperature change.

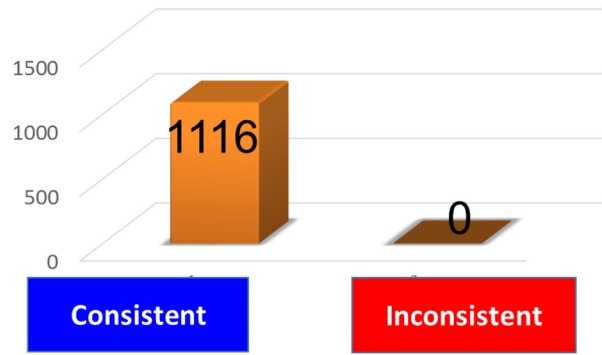

**Figure 9.** Statistics of the longevity test. All chips show no change in failure bit patterns before and after bake.

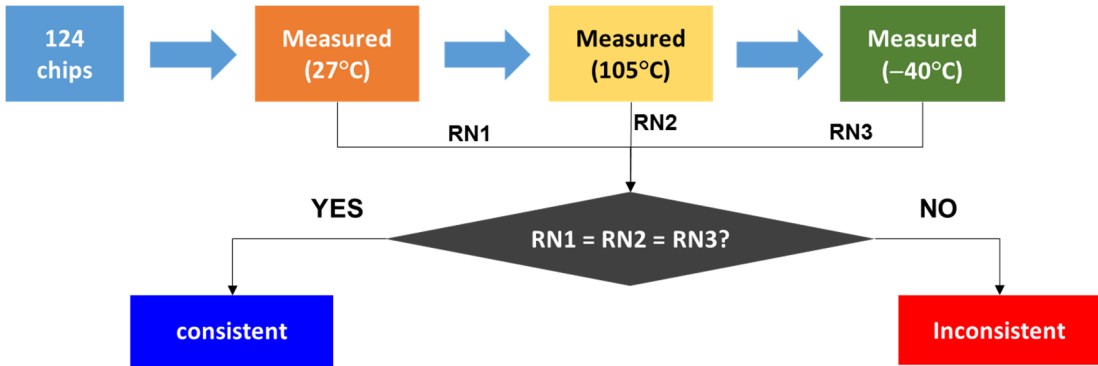

**Figure 10.** A drawing to illustrate temperature the procedure of dependence test. The 124 sample chips are all mass-produced 1Gb DDR1 DRAM IC chips.

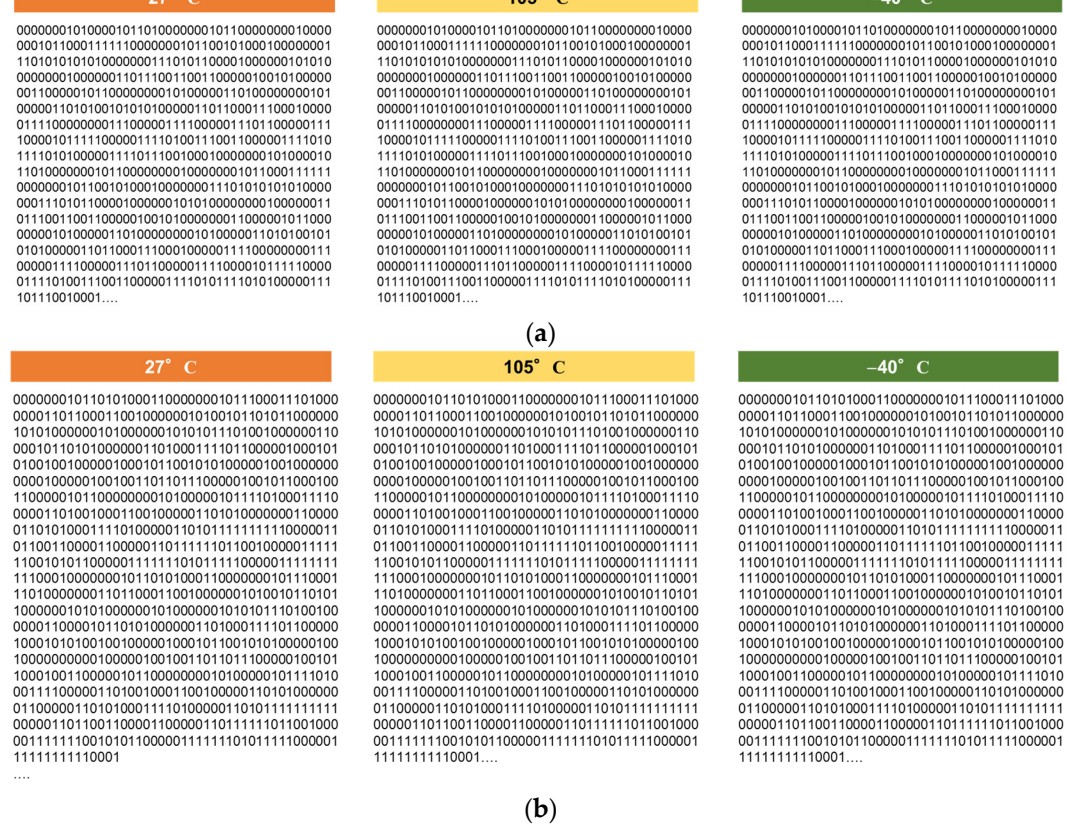

**Figure 11.** A part of the cyber-physical chip identifications measured at 27 °C, 105 °C and −40 °C from the left to right of chip number 16 (**a**) and 121 (**b**) among 124 sample chips. All bits are consistent at all temperatures, respectively. This shows that our cyber-physical chip identification must be resistant to temperature change during the operation.

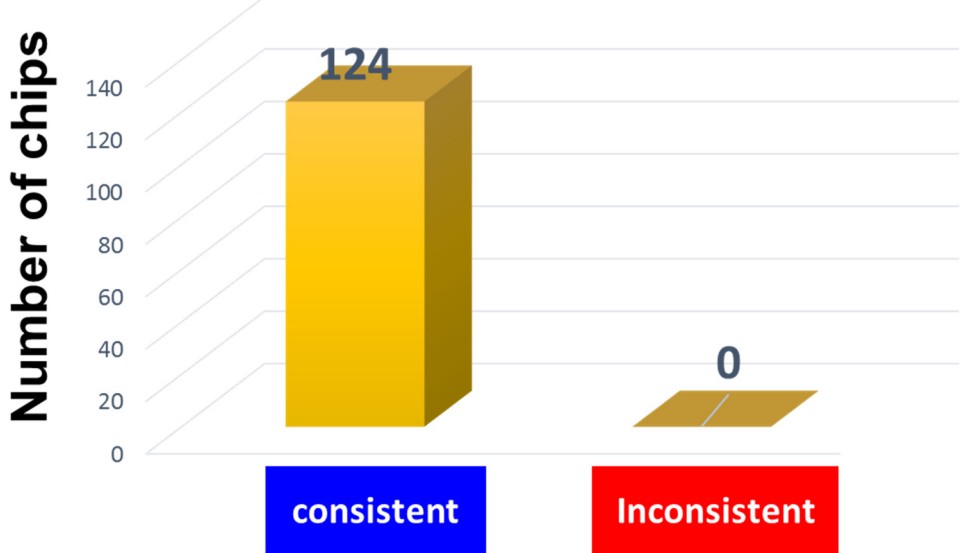

**Figure 12.** Statistics of the temperature test. All chips have no change in the failure bit patterns of cyber-physical chip identification in temperature range from −40 °C to 105 °C.

*3.2. Implementation of Cyber-Phsyical Chip Identification to Blockchain*

If a physical address is uniquely linked to a secret key used in blockchain, this physical address can be uniquely connected to a logical address via the pair of the secret key and the public key, as

illustrated in Figure 13 [28–31]. Therefore, the problem of protection of the data transaction between logical addresses becomes identical to that of the protection between physical addresses. We can install the cyber-physical chip identification to a semiconductor chip included in a connected device by extracting the physical randomness, which is specific to the chip with no change in the application software on the blockchain infrastructure. We can extract any data from a semiconductor chip by changing nothing above the data link layer in Figure 1a–d. In general, upper layer is independent of any change in lower layers of the communication layer structure. Thus, we can insert a new layer below the datalink layer without any change in the blockchain layer, as illustrated in Figure 14 [28–31]. We call this insertion the Cyber-Physical Link (CPL) layer. Note that the logical and physical addresses are uniquely linked, as illustrated in Figure 13 therein. In this way, the data transaction between connected devices is carried out through the CPL layer. As long as data transactions between those logical addresses are protected by blockchain, data transaction between the corresponding physical addresses can be also protected thanks to the CPL layer.

Figure 15a,b illustrates the details of data transaction going through the CPL layer [28]. The top squares are identification cores (ID cores). The bottom ones are transaction units which are identical to those in Figure 2. The ID core (m) is composed of a key generator, a secret key (m), a cyber-physical chip identification code, that is, the CPCID (m), where m is an integer to indicate the node (m). In Figure 15a, the CPCID (m) is an input to a key generator to generate a public key (m) and a secret key (m), which are uniquely coupled according to the algorithm of Rivest, Shamir and Adelman (RSA) [43]. The RSA cryptosystem assumes that it is practically difficult to factorize a large integer. In Figure 15b, the CPCID (m) is hashed to be a secrete key (m). The secret key (m) is then input to a key generator to generate a public key (m) which makes a unique pair with the secret key (m) according to the Elgamal's algorithm [44]. The Elgamal's algorithm and its expansions [45,46] assume that it is practically difficult to compute discrete logarithms. Both problems of factorization and discrete logarithms have been extensively investigated and then the bit length of solved problems has increased annually. As a result, the factorization of a 768-bit length integer was solved in 2010 [47]. The discrete logarithms problem of 923-bit length was solved in 2012 [48]. The encryption algorithm has also been improved simultaneously but if the bit length is less than one kilobits, the possibility that an adversary can obtain the secret key from the public key is non-negligible. It is thus regarded that a couple of kilobits is necessary to protect the secret keys from the advanced cyber-attacks.

We can obtain a unique pair of secret and public keys in each node; and at last each secret key is uniquely linked to a mass-produced memory chip by using the cyber-physical chip identification that can be generated from the failure bit distribution in an on-chip memory cell array. We may appropriately confine each secrete key inside a memory chip while exchanging digital information (data) with other nodes on the network. In addition, in case we remove CPCIDs (i.e., cyber-physical chip identification codes) and key generators from ID cores, Figure 15a,b become identical to Figure 2. The installation of the CPL layer is, therefore, fully compatible to the data transaction going through blockchain. This is a feasible way to implement blockchained IoT.

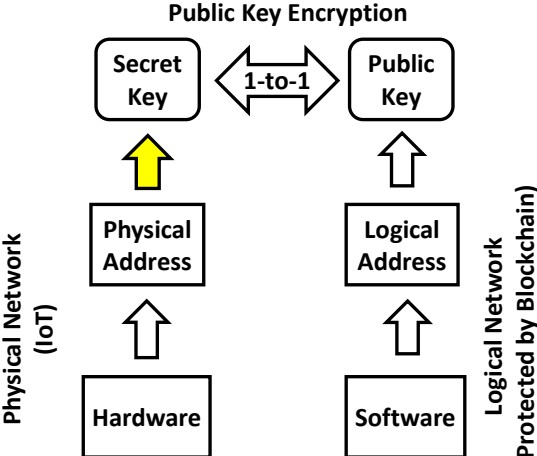

**Figure 13.** Linked physical and logical addresses. A connected device (hardware) and an account in application service (software) are strictly connected by generating a pair of public and secret keys from the cyber-physical chip identification of a chip equipped into the connected device.

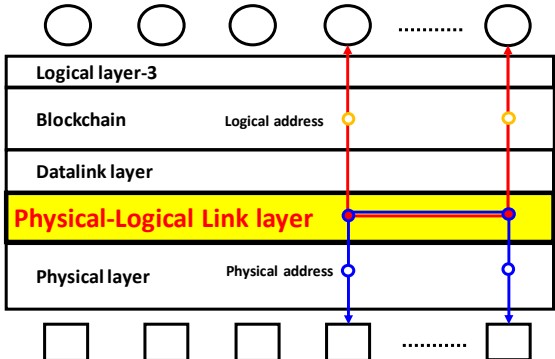

**Figure 14.** Communication layer diagram including the Cyber-Physical Link layer. The usage of cyber-physical chip identification of a chip in a connected device is equivalent to the installation of the physical-logical link layer.

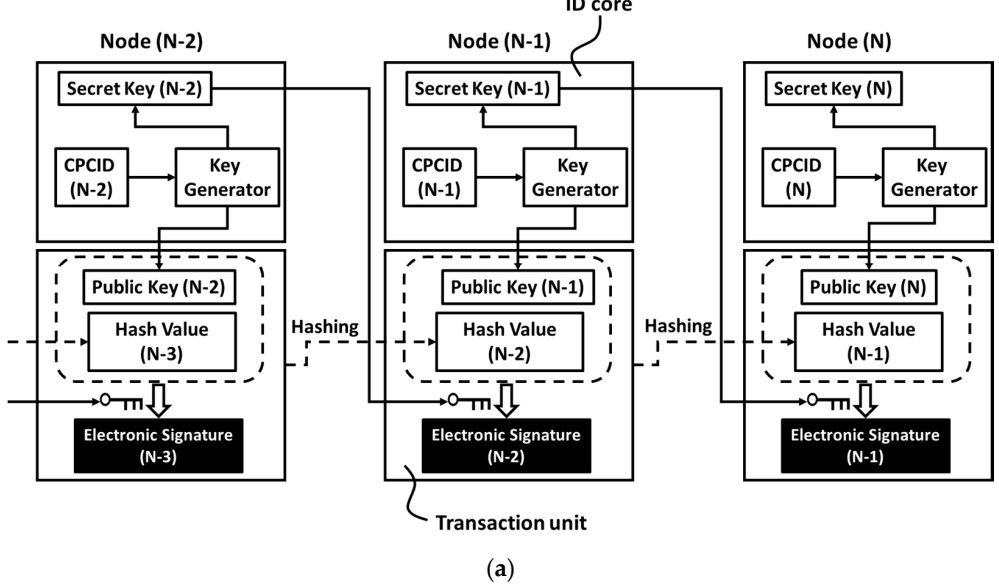

(**a**)

**Figure 15.** *Cont.*

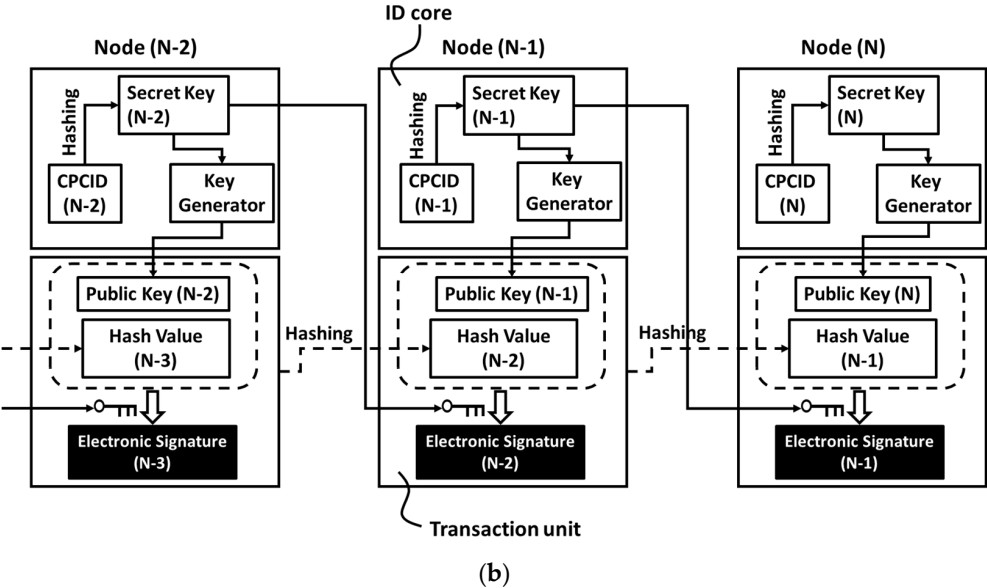

**Figure 15.** Data transaction going through the Cyber-Physical Link layer, wherein the dashed arrow denotes the hashing: (**a**) by using the RSA algorithm and (**b**) by using the Elgamal-type algorithm. CPCID (N−2, N−1 and N) denotes the cyber-physical chip identification codes of the nodes (N−2, N−1 and N), respectively. The usage of cyber-physical chip identification of a chip in a connected device is equivalent to the installation of the ID core in data transaction, the record of which can be protected by blockchain.

## 4. Discussion

As illustrated in Figure 14, the CPL layer is inserted below the datalink layer. Since the upper layers are independent of any change in the lower layers in the communication layer structure, the insertion of the CPL layer will not change any functions of blockchain and will not affect any of the blockchain application services. The cyber-physical chip identification is necessary to form the CPL layer by uniquely linking physical and logical addresses in the IoT network (i.e., cyber-physical network), as illustrated in Figure 13. In order to satisfy the requirement of scalability of blockchain in the IoT network, the cyber-physical chip identification has to be implemented by a mass-produced semiconductor chip that is used in the majority of connected devices in the IoT network. Otherwise, we cannot ensure the protection of the sufficient number of IoT devices. In this work, we have successfully demonstrated the cyber-physical chip identification by a mass-produced DRAM IC chip. Thus, the cyber-physical chip identification can be implemented in 5G base stations or other communication devices to protect massive quantity of data transaction via those communication devices using blockchain. Those communication devices (incl. 5G base stations) may include memory chips like DRAM. Furthermore, 5G base stations commonly experience a self-heating issue due to high-speed communication functionality. The cyber-physical chip identification that is resistant to temperature change in the mass-product level is therefore helpful to protect massive communication among a large number of 5G base stations through blockchain. We have successfully demonstrated that the cyber-physical chip identifications generated from mass-produced DRAM IC chips are unaffected by temperature change.

It is possible to implement this cyber-physical chip identification on all connected devices that have a semiconductor IC chip. Such an IC chip may be either a stand-alone chip of or a system-on chip with embedded volatile or non-volatile memory, for example, DRAM, MRAM, ReRAM, PCRAM, FRAM, NOR Flash, NAND Flash, Mask ROM, Junction ROM, EPROM, EEPROM, OTP and so forth. Among them, a stand-alone DRAM IC chip may be a potential solution for connected devices with sufficient power supply [31]. Additionally, DRAM has been and will be extensively used as a main memory in any kind of processor units (CPU, MPU, GPU, GPGPU . . . ), serving as cash memory in

controllers and communication devices (including 5G base stations), being utilized in edge computing and so forth. Additionally, the stable shipment in large quantity of DRAM IC chip has been guaranteed as it has been used extensively from long ago. This is decisively indispensable to cover as many IoT devices as possible. For example, the application of cash memory (1Gb DDR1 DRAM) to an SSD controller has been demonstrated with no change in front-end-process and no penalty of chip area [29]. Thereby, blockchain can protect data transaction among connected devices with SSD controllers. In a similar way, as we already mentioned, blockchain can be further extended to protect data transaction among any connected devices with any kind of semiconductor IC chips. For example, we can protect data transaction between a connected device with DRAM and another connected device with MRAM.

The registration of a new block to blockchain follows a consensus algorithm, as explained in Section 2.2 by using an example of PoW. On the other hand, the installment of the CPL layer does not affect the consensus mechanism at all and does not put any restrictions on the type of consensus algorithm in use. Thus, we can replace PoW with another consensus algorithm (PoS, PoI, PoC or some new ones) if necessary. Furthermore, any other additional functions or services of blockchain (e.g., smart contract, data flow traceability etc.) are available as is, even after the CPL layer installment. Smart contract is helpful to control the formation of connected devices with the smallest risk of the spoofing and fake signals in the formation. The data traceability in the cyber-network is upgraded to the data traceability in the IoT network. The merits and feasibilities will be discussed more in detail elsewhere.

## 5. Conclusions

The IoT is a physical network under adversaries' attack. The physical network comprises a huge number of connected devices with physical addresses, respectively. On the other hand, the data transactions between logical addresses can be protected by blockchain, no matter how big is the number of logical addresses (scalability). By respectively linking physical addresses to logical ones, as illustrated in Figure 14, we can insert the CPL layer below the datalink layer in the communication layer structure so that the blockchain can protect data transactions in the IoT network. This is the concept of Blockchained Internet-of-Things (BIoT). The important feature of the approach is that the cyber-physical chip identification guarantees a stable shipment in large quantity, which satisfies the scalability of the blockchain and IoT. We conclude that a mass-produced DRAM IC chip is a potential candidate to generate the cyber-physical chip identification. Moreover, since DRAM IC chips used in this demonstration can serve as a source of cyber-physical chip identifications without any change in front-end process and any penalty in chip area, they are memory chips that work in computing system and provide cyber-physical chip identifications. This implies a potential that we can exclude a stand-alone security chip from connected devices. Furthermore, this method is resistant to the temperature changes. Temperature independence is an indispensable characteristic to apply the cyber-physical chip identification on 5G base stations that are known for having the self-heating problem.

## 6. Patents

Patents related to this work are owned by 3H Blockchain, Inc.

**Author Contributions:** Conceptualization, H.W.; methodology, H.W.; software, H.W.; validation, H.W.; formal analysis, H.W.; investigation, H.W.; resources, H.W. and H.F.; data curation, H.W.; writing—original draft preparation, H.W.; writing—review and editing, H.W.; visualization, H.W.; supervision, H.W.; project administration, H.W. and H.F.; funding acquisition, H.F.

**Funding:** This research received no external funding.

**Acknowledgments:** The author would like to express their sincere appreciation to T. Hamamoto, Zentel Japan for the collaboration of the measurement.

**Conflicts of Interest:** The authors declare no conflict of interest. The funders had no role in the design of the study; in the collection, analyses or interpretation of data; in the writing of the manuscript or in the decision to publish the results.

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
