# Peer review of "A Novel Chip-Level Blockchain Security Solution for the Internet of Things Networks"

_technologies, doi:10.3390/technologies7010028_

Round 1

Reviewer 1 Report

Authors proposed a security solution that is based on Block-Chain for IoT. The manuscript sound fine technically, but there are some comments that must be applied as follows:

1.      The abstract should be re-written as it doesn’t explain the results of the paper very well. There is only one sentence regarding to the proposed method.  

2.      Section 2 is about the Blockchain, but it’s subsection, Subsection 2.2, is also titles Blockchain!

3.      The proposed should be compared to the related works to prove the quality of the proposed method.

Author Response

Thank you very much for your reviewing our manuscript and giving positive comments and valuable suggestions. We revised the manuscript according to your suggestions. Please check our reply to your comments as follows:

Ø  1. The abstract should be re-written as it doesn’t explain the results of the paper very well. There is only one sentence regarding to the proposed method. 

Yes, we have. We appended to the abstract the brief description of our measurement by using DRAM chip to show that our cyber-physical chip identification is excellently stable to temperature change. And then, we explained that this method is useful in 5G base station. Also, we added “DRAM” and “IC chip” to keywords’ list. Please see the lines 19 – 24 and 26 underlined in the revised manuscript.

Ø  2.      Section 2 is about the Blockchain, but it’s subsection, Subsection 2.2, is also titles Blockchain!

We revised the title of subsection 2.2 to “Generation of Blockchain”.

Ø  3.      The proposed should be compared to the related works to prove the quality of the proposed method.

The other methods regarding Blockchain and IoT don’t focus on how to connect physical address to logical ones, respectively. We pointed out this connection is indispensable to enable Blockchain to protect communication in IoT for the first time. In addition, we pointed out that mass-product DRAM IC chips to be included in IoT devices can connect physical nodes (IoT devices) to logical accounts in Blockchain, respectively. There is probably no prior works concluding that mass-product IC chips can resolve the problem of connection between physical and logical nodes in Blockchained IoT. With this regard, we cannot find a prior work which is sufficiently related to this topic. However, we can find prior works in the field of physical chip identification. We then compare our cyber-physical chip identification to the prior works such as RNG and PUF in Sec. 3 in the previous manuscript. But, as the reviewer-2 suggested, this section is not readable. Then, we re-wrote it in multiple sub-sections according to the suggestion of the reviewer-2. In addition, our measurement result shows that our cyber-physical chip identification is excellently resistant to temperature change. This is a big advantage to the existing chip technologies (PUF) in the application to 5G base station, because PUF has a serious issue of temperature dependence. To emphasize this, we also added the new sentences in the lines 336 – 340 underlined and the references 38 and 39. With this regard, our method is a brand new one, we believe.

Finally, please see the portions underlined, which are revised according to the other reviewers, in the revised manuscript to check these revisions. Two native English speakers made the proof-edit of the revised manuscript according to the suggestion of the reviewer-3. Thus, there are a plenty of minor revisions suggested in the native speakers’ proof-review. However, we didn’t indicate them in the revised manuscript in order not to make the manuscript complicated. And, we summarize the revision of numbering:

Figs. 7 - 12 are added. Figure numbers are revised from Fig. 7 – 9 to 13 – 15. 

References 22 - 24, 38, and 39 are added. Reference numbers are revised from 21 - 34 and 35 – 40 to 24 – 37 and 40 – 45, respectively.

We would believe that this revision would be allowable to be published.

Reviewer 2 Report

===== Brief paper summary=====

The authors proposed a method to prevent adversaries from spoofing physical or hardware addresses of the IoT devices in the blockchain network.

===== Weakness =====

-  The title of the manuscript is vague. The title does not reflect the problem the authors tried to solve in this manuscript.

-  The abstract is not well written. The contribution of this manuscript is not clear from the abstract

- The problem statement is not clearly defined. The motivation of this study is vague.

- A review of the existing research works was not provided. Are there any prior works on this domain? If so, why the existing methods cannot solve the problem?

- Section 3 presented the proposed model. However, this section is very hard to follow. It is not clear what scheme the authors are trying to propose to solve a problem. This section should be written in multiple sub-sections.

- A security analysis of the proposed method must be provided. The result of the analysis should demonstrate that the proposed method can defend against physical address spoofing attacks.

- Implementation, experiment, and evaluation of the proposed method were not provided. Therefore, the feasibility of the proposed method cannot be validated.

Author Response

Thank you very much for your reviewing our manuscript and then giving valuable suggestions. We revised the manuscript according to your suggestions. Please check our reply to your comments as follows:

Ø  The title of the manuscript is vague. The title does not reflect the problem the authors tried to solve in this manuscript.

We agree to your opinion. We revised the title to “Novel Chip-level Blockchain Security solution for the IoT Networks.

Ø  The abstract is not well written. The contribution of this manuscript is not clear from the abstract

To clarify our contribution in the abstract, we substantially revised the manuscript by adding the sentences in the lines 19 - 24. Please see these lines underlined to check if this revision satisfies your request.

Ø  The problem statement is not clearly defined. The motivation of this study is vague.

To clarify the problem statement and motivation, we substantially revised Sec. 3. In particular, please see Sec. 3.1 and the sentences underlined in the lines 336 – 340.

Ø  A review of the existing research works was not provided. Are there any prior works on this domain? If so, why the existing methods cannot solve the problem?

There is no prior work to connect physical node to logical account by using mass-product IC chip. But there is plenty of prior works of chip encryption devices; e.g., RNG as referred in [11-13] and PUF as referred in [29-32] ([32-35] in the revised manuscript). We reviewed them in the new subsection 3.1.1. Also, to clarify the temperature dependence of the prior works, we append the new references [38] and [39] in the revised manuscript. Please see the lines 336 – 340 underlined.

Ø  Section 3 presented the proposed model. However, this section is very hard to follow. It is not clear what scheme the authors are trying to propose to solve a problem. This section should be written in multiple sub-sections.

We substantially revised Sec. 3 according to your suggestion in the form of multiple sub-section. Please see Sec. 3 to check if this revision satisfies your request.

Ø  A security analysis of the proposed method must be provided. The result of the analysis should demonstrate that the proposed method can defend against physical address spoofing attacks.

The aim of this paper is to show a new method to connect a physical node (a connected device including an IC chip) to a logical account in the blockchain infrastructure by using a cyber-physical chip identification which is excellently resistant to temperature change. Since the temperature resistivity is indispensable in the application to 5G base stations, we showed the measurement data that we obtained by using mass-product IC memory chips in the added subsection 3.1.3. We believe that this method would be a potential countermeasure to the spoofing attack of physical addresses. However, a concrete demonstration of the countermeasure to spoofing of connected devices is our next target to be published elsewhere. The topic of this paper is an interdisciplinary between cyber security and IC technology. We approach from the side of IC technology to this topic and then disclose the first measurement and achievement regarding a component device to connect a physical node to a logical account in the blockchain infrastructure. According to your suggestion, we revised the title to “Novel Chip-level Blockchain Security solution for the IoT Networks”.

Ø  Implementation, experiment, and evaluation of the proposed method were not provided. Therefore, the feasibility of the proposed method cannot be validated.

In the revised manuscript, we append more data in the newly added subsection 3.1.3 other than in Fig. 6 in the previous manuscript. Please see the added figures 7 – 12 and their related sentences in Sec. 3.1.3. By adding these figures, we updated the figure number from previous Figs. 7 – 9 to the revised ones 13 – 15, respectively.

Finally, please see the portions underlined, which are revised according to the other reviewers, in the revised manuscript to check these revisions. Two native English speakers made the proof-edit of the revised manuscript according to the suggestion of the reviewer-3. Thus, there are a plenty of minor revisions suggested in the native speakers’ proof-review. However, we didn’t indicate them in the revised manuscript in order not to make the manuscript complicated. And, we summarize the revision of numbering:

Figs. 7 - 12 are added. Figure numbers are revised from Fig. 7 – 9 to 13 – 15. 

References 22 - 24, 38, and 39 are added. Reference numbers are revised from 21 - 34 and 35 – 40 to 24 – 37 and 40 – 45, respectively.

We would believe that this revision would be allowable to be published.

Reviewer 3 Report

The authors propose a conceptual approach to ensuring security in IoT environments using blockchain (BIoT). The approach combines hardware and software and introduces what they announce as a new concept to prevent attacks. 

I believe that your work, here described, could be enhanced if you address the following issues:

*** the written english of the paper, although is understandable is not accurate and could benefit of an extensive rewriting and proof-editing. 

*** the typos, sentence correction and grammar issues include but are not limited to the following:

- "A Concrete Way to Make Blockchained IoT Real" >> concrete? consider replacing the word with "Actual or Real or..."

- "But the 12  security of IoT is still an open problem."It is supposed that there may be a good chemistry coming13 from these new comers." >> rewrite

- "Enabling blockchain to protect IoT is not 16 self-evident without respectively identifying logical and physical nodes." >> rewrite the self-evident part

- 22 " change the game rule of industrial eco-system" >> change the rules of the game?

- "Most of end users may not 31  manage all of them securely." >> most end-user may not manage all of them in a secure way?

- "For example, a 57 cyber-attack may be succeeded somewhere at last even though">> rewrite

- "By this way, we 106 are required to link physical nodes having the physical substance to logical nodes having" >> "By this way" ?

- "By this way, the logical layers have been repeatedly laminated above the 125  datalink layer, every when the security system is updated to fix the vulnerabilities." >> rewrite

- "Proof-of-State 182  (PoW)," >> PoS?

- "The PoC is adopted in NEM. The PoC is 184 adopted in Ripple." >> PoI?

Regarding the structure you could try to include some extra information in the images that will allow the description text to be more easy to read.

* For instance, in Figure 1 consider adding  extra information regarding the fact that the new logical layers appeared as security layers because the previous one was compromised

* In section "2.1. Transaction" try to rewrite the text -  removing al l the (N) and (N-2) (N-1)... and so on... the text, as it is is not readable

* Providing a field study to actually show that your approach is not merely conceptual - an actual example.

The topic Security in IoT is not new and neither the approach to use blockchain in the solution. You should have more related work depicted so that you can easily prove the novelty that your approach is bringing to science.

* Consider reading and including the following and other sources (as references if appropriate):

- Dorri, Ali, Salil S. Kanhere, and Raja Jurdak. "Blockchain in internet of things: challenges and solutions." arXiv preprint arXiv:1608.05187 (2016).

- Dorri, Ali, Salil S. Kanhere, and Raja Jurdak. "Towards an optimized blockchain for IoT." Proceedings of the Second International Conference on Internet-of-Things Design and Implementation. ACM, 2017.

- Banerjee, Mandrita, Junghee Lee, and Kim-Kwang Raymond Choo. "A blockchain future for internet of things security: a position paper." Digital Communications and Networks 4.3 (2018): 149-160.

- Khan, Minhaj Ahmad, and Khaled Salah. "IoT security: Review, blockchain solutions, and open challenges." Future Generation Computer Systems 82 (2018): 395-411.

- Hammi, Mohamed Tahar, et al. "Bubbles of Trust: A decentralized blockchain-based authentication system for IoT." Computers & Security 78 (2018): 126-142.

- Kouicem, Djamel Eddine, Abdelmadjid Bouabdallah, and Hicham Lakhlef. "Internet of things security: A top-down survey." Computer Networks (2018).

Author Response

Thank you very much for your reviewing our manuscript and giving positive comments and valuable suggestions. We revised the manuscript according to your suggestions. Please check our reply to your comments as follows:

Ø  The written English of the paper, although is understandable is not accurate and could benefit of an extensive rewriting and proof-editing.

According to your suggestion, two native English speakers made the proof-editing of the revised manuscript. After that, we made minor revisions to make sure the readability again.

Ø  the typos, sentence correction and grammar issues include but are not limited to the following:

-          "A Concrete Way to Make Blockchained IoT Real" >> concrete? consider replacing the word with "Actual or Real or..."

We revised the title according to the reviewer-2 to “Novel Chip-level Blockchain Security solution for the IoT Networks”. Then, we believe we have resolved this.

-            "But the 12  security of IoT is still an open problem."It is supposed that there may be a good chemistry coming13 from these new comers." >> rewrite

We revised to “The security of IoT is still an open problem, and if blockchain can reinforce IoT security, as many authors have hoped in recent papers, these new comers appear to be a good much.”  Please see the lines 12 – 14 underlined in the revised manuscript.

-            "Enabling blockchain to protect IoT is not 16 self-evident without respectively identifying logical and physical nodes." >> rewrite the self-evident part

Yes. We revised it to “cannot be brought to reality” according to a native English speaker’s suggestion. Please see the line 17 underlined.

-            22 " change the game rule of industrial eco-system" >> change the rules of the game?

Yes. We revised it according to your suggestion. Please see the line 29 underlined.

-          "Most of end users may not 31  manage all of them securely." >> most end-user may not manage all of them in a secure way?

Yes. We revised it according to your suggestion. Please see the line 38 underlined.

-          "For example, a 57 cyber-attack may be succeeded somewhere at last even though">> rewrite

Yes. We revised it to “This implies that a cyber-attack may succeed somewhere at last even though the probability of an adversary winning each time is less than 0.001%.” However, in this revision, there is also minor revision suggested by a native English speaker, which is not-underlined. Please see the lines 64 - 65 in the revised manuscript.

-          "By this way, we 106 are required to link physical nodes having the physical substance to logical nodes having" >> "By this way" ?

Yes. We revised it to “Therefore”. Please see the line 117 in the revised manuscript.

-          "By this way, the logical layers have been repeatedly laminated above the 125  datalink layer, every when the security system is updated to fix the vulnerabilities." >> rewrite

Yes. We revised it to “Consequently, the logical layers have been repeatedly updated above the datalink layer to fix the vulnerabilities.” However, according to the suggestion of a native English speaker, we revised “laminated” to “updated”. Please see the line 136 – 137 in the revised manuscript.

-          "Proof-of-State 182  (PoW)," >> PoS?

Yes. We revised it according to your suggestion. Please see the line 194 in the revised manuscript.

-          "The PoC is adopted in NEM. The PoC is 184 adopted in Ripple." >> PoI?

Yes. We revised it according to your suggestion. Please see the line 196 in the revised manuscript.

Ø  Regarding the structure you could try to include some extra information in the images that will allow the description text to be more easy to read.

-            For instance, in Figure 1 consider adding  extra information regarding the fact that the new logical layers appeared as security layers because the previous one was compromised

Yes. According to your suggestion:

we append the sentence “Each new layer appear as security layers because the previous one was compromised.” to the end of the caption of Fig. 1.  However, in this revision, there is also minor revision suggested by a native English speaker, which is not-underlined.

We also added the sentence of “This was extracted from a mass-produced memory chip of 1Gb DDR1 DRAM. We can regard that in fact different chips exhibit unequal cyber-physical chip identification codes.” to the end of caption of Fig. 6.

We also added the sentence of “A connected device (hardware) and an account in application service (software) are strictly connected by generating a pair of public and secret keys from the cyber-physical chip identification of a chip equipped into the connected device.” to the end of the caption of the revised Fig. 13 (previous Fig. 7).

We also added the sentence of “The usage of cyber-physical chip identification of a chip in a connected device is equivalent to the installation of the physical-logical link layer.” to the end of the caption of the revised Fig. 14 (previous Fig. 8).

We also added the sentence of “The usage of cyber-physical chip identification of a chip in a connected device is equivalent to the installation of the ID core in data transaction, the record of which can be protected by blockchain.” to the end of the caption of the revised Fig. 15 (previous Fig. 9).

-            In section "2.1. Transaction" try to rewrite the text -  removing al l the (N) and (N-2) (N-1)... and so on... the text, as it is not readable

Yes. We mandatory revised Sec. 2.1 according to your request. Please see the revised Sec. 2.1 to check if this revision satisfies your request.

-            Providing a field study to actually show that your approach is not merely conceptual - an actual example.

Thank you very much for your valuable suggestion. Since the aim of this paper is to propose a new chip-level solution to connect physical node to logical account in Blockchain infrastructure. This is a study of this domain from the view point of LSI technology. Then, we haven’t performed the field study yet. However, your suggestion should be very helpful to our coming activity. Please let us leave it to our future study.

Ø  The topic Security in IoT is not new and neither the approach to use blockchain in the solution. You should have more related work depicted so that you can easily prove the novelty that your approach is bringing to science.

* Consider reading and including the following and other sources (as references if appropriate):

Thank you very much for informing us these valuable references. Since these are articles regarding Blockchain and IoT, we added the three references (Dori, Mandrita, and Kouichem) that we can retrieve from the journal web sites to after [21] and before [22] in the previous manuscript because [16 – 21] are also the references related to BIoT. By this addition, the reference numbers are revised, too.

Finally, please see the portions underlined, which are revised according to the other reviewers, in the revised manuscript to check these revisions. As a result of the proof-editing by two native English speakers, there are a plenty of minor revisions suggested in the native speakers’ proof-review. However, we didn’t indicate them in the revised manuscript in order not to make the manuscript complicated. And, we summarize the revision of numbering:

Figs. 7 - 12 are added. Figure numbers are revised from Fig. 7 – 9 to 13 – 15. 

References 22 - 24, 38, and 39 are added. Reference numbers are revised from 21 - 34 and 35 – 40 to 24 – 37 and 40 – 45, respectively.

We would believe that this revision would be allowable to be published.

Round 2

Reviewer 2 Report

- An ariticle is missing in the title of the paper. The title should be revised as: A Novel Chip-level Blockchain Security Solution for the  IoT Networks

- The authors are advised to increase the number of reference with recent research works

- The authors adequately extended the manuscript with a better problem statement, motivation of the work, background, and experimental details

Author Response

Thank you very much for your careful and valuable suggestions and comments. Due to your support, we have completed the manuscript. Please check if we adequately satisfy your suggestions.

²  An ariticle is missing in the title of the paper. The title should be revised as: A Novel Chip-level Blockchain Security Solution for the IoT Networks

Yes. We revised the title by adding “A” at the beginning.

²  The authors are advised to increase the number of reference with recent research works

We added the references 25 – 27 according to your advice. Thus, the reference numbers 25 - 45 are revised to 28 – 48, respectively. Moreover, in order to acquire further readability, we added the sentence of “How can semiconductor companies…in the IoT network”, using the new [26]. Please see the portions underlined in the lines 122 – 125.

²  The authors adequately extended the manuscript with a better problem statement, motivation of the work, background, and experimental details

Thank you very much. This is a result by your help.

Finally, please see the portions underlined, which are revised according to the reviewer-3, in the revised manuscript.

We believe that this revision would be allowable to be published.

Reviewer 3 Report

Your paper has gone to a serious improvement. Congratulations on achieving this work.

Nevertheless, I could still find a couple of typos:

Add "A" to the beginning of the title.

"The security 14 of IoT is still an open problem, and if blockchain can reinforce IoT security, as many authors have 15 hoped in recent papers, these new comers appear to be a good much." >> I don't understand what you are trying to say with "these new comers appear to be a good much.". it should be newcomers. And they appear to be a good much what?

"Each new layer 158 appear as security layer because the previous one was compromised" >> "Each new layer 158 appears as security layer because the previous one was compromised

References 38 and 39 seem to be duplicates.

Author Response

²  Your paper has gone to a serious improvement. Congratulations on achieving this work. Nevertheless, I could still find a couple of typos:

Thank you very much for your careful and valuable suggestions and comments. Due to your support, we have completed the manuscript. Please check if we adequately satisfy your suggestions.

²  Add "A" to the beginning of the title.

Yes. We revised the title by adding “A” at the beginning.

²  "The security 14 of IoT is still an open problem, and if blockchain can reinforce IoT security, as many authors have 15 hoped in recent papers, these new comers appear to be a good much." >> I don't understand what you are trying to say with "these new comers appear to be a good much.". it should be newcomers. And they appear to be a good much what?

We revised this sentence again according to your suggestion to: “these newcomers appear to make a good collaboration to reinforce IoT security.” Please see the portions underlined in the lines 15 – 16 in the revised manuscript and check if this revision can satisfy your request.

²  "Each new layer 158 appear as security layer because the previous one was compromised" >> "Each new layer 158 appears as security layer because the previous one was compromised

Yes. We added “s” to the end of “appear”. Please see the portion underlined in the line 162 and check if this revision can satisfy your request.

²  References 38 and 39 seem to be duplicates.

Thank you very much for informing this. The references [38] (revised to [41]) and [39] (revised to [42]) are from the same group and independent patents having different patent publication numbers JP5857726B2 and JP5870675B2, respectively. However, we revised the order of the 4-th and 5-th inventors in [38] (revised to [41]) because we have found the mistake thanks to your comment. Please see the portion underlined in the line 673. Also, we inserted [38] (revised to [41]) in the line 344. Please see the portion underlined in that line.

Finally, by adding references 25 – 27 according to the reviewer-2’s advice, the reference numbers are revised from 25 - 45 to 28 – 48, respectively. Also, please see the portions underlined, which are revised according to the reviewer-2, in the revised manuscript.

We believe that this revision would be allowable to be published.